# Cystic Fibrosis Cases Missed by Newborn Bloodspot Screening—Towards a Consistent Definition and Data Acquisition

**DOI:** 10.3390/ijns9040065

**Published:** 2023-11-21

**Authors:** Anne Munck, Kevin W. Southern, Jared Murphy, Karin M. de Winter-de Groot, Silvia Gartner, Bülent Karadag, Nataliya Kashirskaya, Barry Linnane, Marijke Proesmans, Dorota Sands, Olaf Sommerburg, Carlo Castellani, Jürg Barben

**Affiliations:** 1Hospital Necker Enfants-Malades, AP-HP, CF Centre, University Paris Descartes, 75015 Paris, France; anne.munck1@gmail.com; 2Department of Women’s and Children’s Health, University of Liverpool, Liverpool L69 3BX, UK; kwsouth@liv.ac.uk (K.W.S.);; 3Department of Paediatric Pulmonology & Allergology, Wilhelmina Children’s Hospital, University Medical Centre Utrecht, Utrecht University, 3508 AB Utrecht, The Netherlands; k.m.dewinter@umcutrecht.nl; 4Pediatric Pulmonology and Cystic Fibrosis Unit, Hospital Universitari Vall d’Hebron, 08035 Barcelona, Spain; silvia.gartner@vallhebron.cat; 5Department of Pediatric Pulmonology, Marmara University, 34890 Istanbul, Turkey; bkaradag@hotmail.com; 6Laboratory of Genetic Epidemiology, Research Centre for Medical Genetics, Moscow Regional Research and Clinical Institute, Moscow 115522, Russia; nataliya.kashirskaya1963@gmail.com; 7School of Medicine and Centre for Interventions in Infection, Inflammation & Immunity (4i), University of Limerick, V94 T9PX Limerick, Ireland; barry.linnane@hse.ie; 8Division of Woman and Child, Department of Pediatrics, University Hospitals Leuven, 3000 Leuven, Belgium; marijke.proesmans@uzleuven.be; 9Cystic Fibrosis Department, Institute of Mother and Child, 01-211 Warsaw, Poland; dorsan8@gmail.com; 10Paediatric Pulmonology, Allergology & CF Centre, Department of Paediatrics III, University Hospital Heidelberg, 69120 Heidelberg, Germany; olaf.sommerburg@med.uni-heidelberg.de; 11Translational Lung Research Center, German Lung Research Center, University Hospital Heidelberg, 69120 Heidelberg, Germany; 12Cystic Fibrosis Center, IRCCS Istituto Giannina Gaslini, 16147 Genova, Italy; carlocastellani@gaslini.org; 13Paediatric Pulmonology & CF Centre, Children’s Hospital of Eastern Switzerland, 9000 St. Gallen, Switzerland

**Keywords:** cystic fibrosis, newborn screening, evaluation, sensitivity, false negatives, missed cases

## Abstract

Repeated European surveys of newborn bloodspot screening (NBS) have shown varied strategies for collecting missed cases, and information on data collection differs among countries/regions, hampering data comparison. The ECFS Neonatal Screening Working Group defined missed cases by NBS as either false negatives, protocol-related, concerning analytical issues, or non-protocol-related, concerning pre- and post-analytical issues. A questionnaire has been designed and sent to all key workers identified in each NBS programme to assess the feasibility of collecting data on missed cases, the stage of the NBS programme when the system failed, and individual patient data on each missed case.

## 1. Background

The recent European survey of the newborn bloodspot screening (NBS) for cystic fibrosis (CF) [1] demonstrated that the framework of parameters established by the European CF Society (ECFS) Neonatal Screening Working Group (NSWG) [2] can enable a more valid comparison of protocol performance. However, there is still room for improvement, as high-quality data depends on continuous data collection, preferably through a centrally coordinated system. In this survey, only 75% of all national programmes achieved the aim of the minimum sensitivity of 95% according to the European best practice guideline [3,4]. A lower sensitivity was found in countries not including DNA analysis in their algorithm (*n* = 7, mean 90%, 95% CI: 80–100%) compared to those that used DNA panels (*n* = 10, mean 95%, 95% CI: 90–100%) or extended genome analysis (EGA, *n* = 4, mean 97%, 95% CI: 95–100%). However, the observation time for missed cases was only 2 years, and we do not know which criteria were used for missed cases in the different countries.

The accuracy of the sensitivity evaluation depends on the time interval of collection as well as the reliability of recording missed cases. Children with a CF screen-positive, inconclusive diagnosis (CFSPID) designation are not included in the evaluation of CF missed cases. To date, there is no uniformly accepted, standardised definition of what is meant by false negative results and over what time period they should be recorded. Rock et al. have listed many factors accounting for CF cases missed by NBS [5]. However, strategies for collecting missed cases differ among countries or regions; information on data collection is often missing, and the duration of follow-up since birth is often too short or not specified.

Since 2000, there have been 15 surveys where individual patient data were reported for 138 missed cases (we excluded cases with meconium ileus as CF diagnosis is not delayed) following a negative NBS result [6,7,8,9,10,11,12,13,14,15,16,17,18,19,20,21] (Table 1). Among these missed cases, 90.6% (125/138) were protocol-related, and 73.9% (102/138) had the first immunoreactive trypsinogen value (IRT-1) below the cut-off. In Europe, only a few countries reported their age range at diagnosis, demographics, symptoms, factors accounting for missed cases, and their *CFTR* variants.

We, therefore, aimed to establish a standardised definition of cases missed by NBS in order to prepare a European survey evaluating the current strategies implemented for collecting and characterising these cases.

## 2. Methods

A core panel of experts (A.M., J.B., K.W.S., and C.C.) asked the ECFS NSWG core committee to provide their current terminology and definitions used in the country. Subsequently, in online video conferences and on-site meetings, we agreed on a unified terminology and definition. Simultaneously, the core panel worked on a questionnaire, with refinements provided following the discussion within the core committee. The questionnaire will be sent to the key workers identified in each NBS programme to assess the feasibility of collecting a panel of data on CF missed cases.

## 3. Results

Terminology and definitions provided by six core committee members from England, Italy, France, Ireland, the Netherlands, and Switzerland were diverse, i.e., “missed cases”, “affected not detected”, “undetected cases” and according to when it happens “true-false negative”, “false negative protocol related case”, “false negative non-protocol related case”, “unidentified case”, and “missed cases”. The following harmonised definition was then established:Missed cases are children and adolescents with a diagnosis of CF—pancreas insufficient (PI) or sufficient (PS)—who were not detected by the NBS programme. There are three stages of the NBS programme when this can happen: pre-analytic, analytic, or post-analytic (Table 2).Missed cases are divided into the following:
(1)false negative cases NBS protocol-related (analytical issues) and(2)false negative cases NBS non-protocol-related (pre- and post-analytical issues)
Infants with meconium ileus (MI) diagnosed with CF shortly after birth, who have a false negative NBS result, need to be reported but will be analysed separately.Sensitivity is calculated from the total number of missed cases (Groups 1 and 2 above), both including and not including those with MI.For quality improvement of NBS programmes, separate analysis should be undertaken using NBS protocol-related (analytical issues) or non-protocol-related (pre-post-analytical issues) results to better identify the underlying issues with the programme.

The newly produced questionnaire “Current strategies for collecting CF cases missed by NBS” (Appendix A) includes general information on who completed the survey, eleven questions focusing on strategies to identify missed cases diagnosed with CF in 2023, including the structure in place to ensure exhaustive reporting of the cohort, a list of items collected at diagnosis for these cases, the stage of the NBS programme when this occurs, and an agreement to complete later on an individual questionnaire for all missed cases reported.

## 4. Discussion

Sensitivity is a key parameter to determine the effectiveness of an NBS programme and an important metric for the ECFS standards. Before any changes are made in a NBS algorithm to improve sensitivity (or specificity), good-quality data over a longer period of time is needed. Regarding missed cases, the distinction between pre-analytical, analytical, and post-analytical problems is important because some do not concern analytical issues or the algorithm in the laboratory, respectively.

For example, pre-analytical issues like a dried blood sample (DBS) not being taken, incorrectly labelled, or taken on the wrong day should not be mixed up with real analytical issues like an IRT value below the cut-off or variants not identified in DNA analysis (Table 2). The same is true for post-analytical issues like administrative error in reporting the NBS result to the CF team, miscommunication of NBS results between primary care provider and family, or error in measurement of sweat chloride.

Recently, an issue has arisen with the use of modulator therapy elexacaftor-tezacaftor-ivacaftor (ETI) during pregnancy. This can lead to a normal NBS results in an infant with CF as, following in utero exposure, the IRT level is below the cut-off [22]. This pre-analytical problem cannot be addressed by changing the algorithm in the laboratory, but requires a different approach, that all children of mothers treated with ETI should always receive a sweat test and/or genetic analysis. More data are needed, as at the moment it is uncertain if these cases should be analysed separately for infants with meconium ileus.

Following a robust and inclusive process, we have established a clear definition for missed cases following a negative NBS result for CF. We have also clarified how these cases should inform sensitivity analysis. It is necessary to have reliable and consistent data collected in the same way across all programmes to compare performance, first against ECFS standards and also between different screening approaches. The reliability of the sensitivity calculation depends on the duration of the acquisition of data on the missed cases and the definition of these cases. Mechanisms (preferably centralised) should be in place for the continuous collection of cases missed by NBS to provide complete data acquisition and data quality for each programme.

## Figures and Tables

**Table 1 IJNS-09-00065-t001:** Clinical and laboratory characteristics of missed cases from the literature (*n*, 138).

Number	Total Cases (*n* = 138)	Protocols Not Using DNA Analysis (*n* = 54)	Protocols Using DNA Analysis (*n* = 84)
Age at diagnosis in months, *n* (%) Median [Q1–Q3] (range)	132/138 (96) 11 [4–36.5] (1–264)	54/54 (100) 11 [4.5–60] (1–264)	78/84 (93) 8.5 [4–24] (1–180)
Diagnosis <3 years, *n* (%)	102/132 (77)	35/54 (65)	67/78 (86)
Symptoms or family history, *n* (%) *	110/138 (80)	50/54 (93)	60/84 (71)
Respiratory	66/110 (60)	27/50 (54)	39/60 (65)
Failure to thrive	48/110 (44)	14/50 (28)	34/60 (57)
Gastrointestinal	24/110 (22)	12/50 (24)	12/60 (20)
Dehydration	19/110 (17)	12/50 (24)	7/60 (12)
Sibling	10/110 (9)	5/50 (10)	5/60 (8)
Chronic rhinosinusitis	5/110 (5)	3/50 (6)	2/60 (3)
Other: pancreatitis, finger clubbing, prolapsus, salted skin, neonatal cholestasis	4/110 (4)	2/50 (4)	2/60 (3)
Sweat test result, *n* (%)	100/138 (72)	48/54 (89)	52/84 (62)
≥60 mmol/L or positive, *n* (%)	88/100 (88)	42/48 (87)	46/52 (89)
<60 mmol/L total, *n* (%) <60 mmol/L (with two CF causing variants), *n* (%)	12/100 (12) 4/12 (33)	6/48 (13) 2/6 (33)	6/52 (12) 2/6 (33)
Allele identification, *n* (%)	219/276 (79)	88/276 (32)	131/276 (47)
Pancreatic status reported	108/138 (78)	54/54 (100)	54/84 (64)
Pancreatic insufficiency *n* (%) **	47/108 (44)	16/54 (30)	31/54 (57)
Reason for missed cases	138/138 (100)	54/54 (100)	84/84 (100)
Dry blood spot not obtained/lost (*n* (%))	3/138 (2)	0/54 (0)	3/84 (4)
IRT1 < cut-off	102/138 (74)	41/54 (76)	61/84 (73)
IRT2 < cut-off	11/138 (8)	9/54 (17)	2/84 (2)
No variant detected	5/138 (4)	0/54 (0)	5/84 (6)
0 variant and IRT-1 below cut-off for IRT-2	8/138 (6)	0/54 (0)	8/84 (10)
Human error	3/13 (2)	0/54 (0)	3/84 (4)
Error in sweat chloride measurement	6/138 (4)	4/54 (7)	2/84 (2)

* Symptoms not reported in 28 cases, some cases had more than one symptom; ** faecal elastase 1 measurement low, established on pancreatic enzyme replacement therapy, or reported as pancreatic insufficiency. DNA, deoxyribonucleic acid; IRT-1, first immune-reactive trypsinogen, taken in the first week of life; IRT-2, second immune-reactive trypsinogen, taken after 2–3 weeks; Q, Quartile.

**Table 2 IJNS-09-00065-t002:** Examples of issues that may lead to missed cases.

**Pre-analytic issues:**
1. Dried blood spot (DBS) sample not obtained/lost
2. DBS sample labelling error in the neonatal nursery
3. DBS sample mix-up
4. DBS quality is unacceptable
5. DBS taken on the wrong day (too early/too late)
6. Mother takes elexacaftor-tezacaftor-ivacaftor (ETI) during pregnancy (not reported)
**Analytic issues:**
1. Infant’s first IRT (IRT-1) or IRT-1/PAP level is below the cut-off
2. In IRT/IRT protocol: The second IRT (IRT-2) is below the cut-off
3. In IRT/DNA protocol: Variants of missed cases not in the panel
4. In IRT/DNA protocol: Failure in DNA analysis
5. In IRT/DNA protocol: No detected variant and IRT-1 is below the ultrahigh cut-off for the next step IRT-2
6. In IRT/DNA protocol: No detected variant and ultrahigh IRT-1, next step IRT-2 is below the cut-off
**Post-analytic issues:**
1. IRT-1 or IRT-1/PAP values above cut-offs are not actioned
2. In IRT/IRT protocol: A 2nd specimen is not obtained, no follow-up
3. Administrative error in reporting the NBS result to the CF team/primary care provider/family
4. Miscommunication of NBS results between primary care provider and family (family not reachable, not willing to come for sweat testing)
5. Error in measurement of sweat chloride concentration (SCC)
6. Appropriate SCC cut-off values are not used
7. SCC is normal after detection of only one variant (in IRT/DNA protocol)

## Data Availability

Data are contained within the article and Appendix A.

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
