# Peer review of "Cystic Fibrosis Cases Missed by Newborn Bloodspot Screening—Towards a Consistent Definition and Data Acquisition"

_2409-515X, 2023, doi:10.3390/ijns9040065_

Round 1

Reviewer 1 Report

Comments and Suggestions for Authors

The paper admirably aims to improve and standardize data collection in regard to missed cases of Cystic Fibrosis in Europe in order to gain more knowledge on how to reduce the number of missed cases.

Overall it is a nice paper, but the text could be a bit more cohesive.   

Before the manuscript is ready for publication, there is some issues that need to be addressed.

1.       The authors states in the “result” section, that sensitivity should be calculated as total number of missed cases both including and not including meconium ileus (MI). In the “background” section, the authors list the results of a recent survey and they provide the sensitivity found in different countries with and without DNA analysis. The authors state, that the recommended sensitivity should be no less than 95%. The authors refer to ECFS best practice guideline 2014 and the 2018 revised version. In the 2014 version MI is not addressed, whereas the 2018 version states that sensitivity should be calculated both including and not including MI. It is not clear whether the calculated sensitivity is including or not including MI cases. If the 2018 version of the European guidelines is referenced, the authors should provide both of the sensitivity calculations or at least address why one was not preformed 

2.       The authors reference Rock et al ( reference 5) stating that the paper lists many factors accounting for CF cases missed by NBS. The referenced paper only focuses on infants with a false positive screening result and primarily investigate the correlation between low apgar score and false positive IRT. The authors should consider changing the reference 

3.       Page 2, paragraph 2. The Authors summarize the results of 15 surveys in regard to missed cases. They then state, that in Europe little is known about the demographics and overall factors that account for missing cases. As some of the 15 survey comes from European countries, this statement seems a bit overstated and should perhaps be nuanced. 

4.       In regard to Pre-analytical issues 1. Dried blood spot sample not obtained/lost. It might be reasonable to distinguish between samples not obtained due to errors from heath personnel and samples not obtained because the parents have opted out of newborn screening. The first category is in essence unacceptable errors that needs to be addressed whereas the latter is difficult to address and a category of missed cases that we cannot avoid.

Author Response

We have provided a detailed response-to-reviewer in the attached document. 

Kind regards, 

Jürg Barben 

Reviewer 2 Report

Comments and Suggestions for Authors

This brief report from the ECFS NSWG is very important to improve the collecting of data on missed cases of CF. It represents the results of the process of finding consistent definitions and data acquisition in different countries and for various algorithms. In my opinion, the report is well written and almost ready for publication. I have only few comments on Table 1:

The (n) for total cases should be in the first column and omitted after Total cases in the second

As in all other parts the second line should read: Median age at diagnosis, n (%) and describe the n for the different columns (132/138 54/54 78/84) here and not as “number” in the 4th line

The % figures for the sweat test results are missing

It would be interesting to have some information about children with meconium ileus in this table. At least it should be mentioned whether these cases are included in the reasons for missed cases with IRT 

Author Response

(The authors gave the same response as above.)
